# Estimating global nitrous oxide emissions by lichens and bryophytes with a process-based productivity model

Philipp Porada[1,2], Ulrich Pöschl[3], Axel Kleidon[4], Christian Beer[1,2], and Bettina Weber[3]

[1]Department of Environmental Science and Analytical Chemistry (ACES), Stockholm University, 10691 Stockholm, Sweden
[2]Bolin Centre for Climate Research, Stockholm University, 10691 Stockholm, Sweden
[3]Max Planck Institute for Chemistry, P.O. Box 3060, 55020 Mainz, Germany
[4]Max Planck Institute for Biogeochemistry, P.O. Box 10 01 64, 07701 Jena, Germany

*Correspondence to:* Philipp Porada (philipp.porada@aces.su.se)

**Abstract.** Nitrous oxide is a strong greenhouse gas and atmospheric ozone - depleting agent, which is largely emitted by soils. Recently, lichens and bryophytes have also been shown to release significant amounts of nitrous oxide. This finding relies on ecosystem-scale estimates of net primary productivity of lichens and bryophytes, which are converted to nitrous oxide emissions by empirical relationships between productivity and respiration, as well as between respiration and nitrous oxide release Here we obtain an alternative estimate of nitrous oxide emissions which is based on a global process-based non-vascular vegetation model of lichens and bryophytes. The model quantifies photosynthesis and respiration of lichens and bryophytes directly as a function of environmental conditions, such as light and temperature. Nitrous oxide emissions are then derived from simulated respiration assuming a fixed relationship between the two fluxes. This approach yields a global estimate of 0.27 (0.19 - 0.35) $(\text{Tg N}_2\text{O}) \, \text{yr}^{-1}$ released by lichens and bryophytes. This is lower than previous estimates, but corresponds to about 50 % of the atmospheric deposition of nitrous oxide into the oceans or 25 % of the atmospheric deposition on land. Uncertainty in our simulated estimate results from large variation in emission rates due to both physiological differences between species and spatial heterogeneity of climatic conditions. To constrain our predictions, combined online gas exchange measurements of respiration and nitrous oxide emissions may be helpful.

## 1  Introduction

Lichens and bryophytes have increasingly been recognized to play a relevant role in global biogeochemical cycles (Elbert et al., 2012; Sancho et al., 2016; Barger et al., 2016). They are globally abundant, growing on soils, rocks, and epiphytically on trees. At high latitudes, they may form extensive covers on the forest floor and in wetlands, mosses frequently represent the dominant vegetation type. In drylands, lichens and bryophytes form so-called biological soil crusts together with photosynthesizing cyanobacteria, algae, fungi and bacteria. These crusts cover vast areas in arid and semiarid ecosystems.

In a first approach, based on empirical upscaling of field measurements according to ecosystem categories, Elbert et al. (2012) calculated that lichens and bryophytes, together with free-living cyanobacteria and algae, fix around 14.3 $(\text{Gt CO}_2) \, \text{yr}^{-1}$ (3.9 Gt carbon) at the global scale. This corresponds to about 7 % of the net primary productivity (NPP) by terrestrial vegetation. In an alternative approach to the empirical upscaling of observations, Porada et al. (2013) utilized a process-based non-vascular

vegetation model for lichens and bryophytes, called LiBry, to calculate the NPP of these organism groups at the global scale, obtaining similar results.

In addition to photosynthetic carbon uptake, lichens and bryophytes are able to fix nitrogen through a symbiosis with cyanobacteria (DeLuca et al., 2002; Barger et al., 2016). Together with free-living cyanobacteria, their nitrogen fixation was estimated to sum up to a global value of $\sim$49 $(\mathrm{Tg\,N})\,\mathrm{yr}^{-1}$ (Elbert et al., 2012), which accounts for nearly half of the biological nitrogen fixation on land. The LiBry model yielded a similar estimate of up to 34 $(\mathrm{Tg\,N})\,\mathrm{yr}^{-1}$, based on the nitrogen requirements of lichens and bryophytes determined by Porada et al. (2014). Moreover, it was found in the same study that the organisms may contribute significantly to biotic enhancement of global chemical weathering, by release of weathering agents such as organic acids. Their potential for chemical weathering was derived from their phosphorus demand, assuming that they dissolve surface rocks to acquire phosphorus.

Recently, lichen- and bryophyte-related nitrogen fluxes other than fixation of nitrogen have been shown to be significant at the global scale. Weber et al. (2015) found that biological soil crusts, which may contain large fractions of lichens or bryophytes, emit considerable quantities of the reactive trace gases NO and HONO, accounting for $\sim$1.7 $(\mathrm{Tg\,N})\,\mathrm{yr}^{-1}$. This corresponds to $\sim$20 % of global nitrogen oxide emissions from soils under natural vegetation (Ciais et al., 2013).

Furthermore, Lenhart et al. (2015) showed that a large variety of lichen and bryophyte species release nitrous oxide ($N_2O$). They estimated that the organisms emit a total value of 0.45 (0.32 - 0.59) $(\mathrm{Tg\,N})_2\mathrm{O}\,\mathrm{yr}^{-1}$ at the global scale, which corresponds to 4 - 9 % of natural terrestrial $N_2O$ emissions (Zhuang et al., 2012). Since $N_2O$ is an important greenhouse gas and also the main depleting substance of stratospheric ozone which is still emitted today, quantifying all contributing sources is of high importance (Butterbach-Bahl et al., 2013; Ravishankara et al., 2009; Gärdenäs et al., 2011; Ciais et al., 2013).

Absolute values of $N_2O$ release estimated by Lenhart et al. (2015) were highest for lichens and bryophytes living on the ground in the boreal zone and for epiphytic lichens and bryophytes in the humid tropics. The relative contributions of lichens and bryophytes to total ecosystem $N_2O$ emissions, however, were highest in desert and tundra biomes, due to the low emissions by other vegetation and the soil there. The high relevance of lichens and bryophytes for $N_2O$ emissions in drylands and at high latitudes is in accordance with their strong impacts on other components of the nitrogen cycle in these regions. Bryophytes, for instance, have been suggested to be the main source of nitrogen input into boreal forests through fixation from the atmosphere by cyanobacterial partners (DeLuca et al., 2002). Also in drylands, lichens and bryophytes are crucial for input of nitrogen into the ecosystem (Barger et al., 2016), and they may even be essential providers of nitrogen for vascular plants (Stewart, 1967; Hawkes, 2003).

The estimate by Lenhart et al. (2015) is derived from measuring emissions of $N_2O$ by the organisms in the laboratory under a range of environmental conditions. All lichen and bryophyte species analyzed by Lenhart et al. (2015) showed release of $N_2O$. Lichens and bryophytes were shown to utilize $^{15}$N labelled $NO_3^-$ but not $NH_4^+$, indicating that $N_2O$ is likely formed during denitrification. The exact process of $N_2O$-formation, however, remains largely unknown. One option is that the organisms themselves release $N_2O$ during the metabolisation of nitrate, in a similar way as suggested by Smart and Bloom (2001) for vascular plants. Another option is, that bacteria growing on lichen and moss cushions are responsible for the emissions of $N_2O$. This second option is supported by a recently published study, where several strains of the bacterial genus *Burkholderia*, which

were shown to emit $N_2O$, were isolated from the boreal peat moss *Sphagnum fuscum* (Nie et al., 2015). While Lenhart et al. (2015) describe that the substrate, which the organisms grew on, was thoroughly removed, further cleaning steps to remove potential bacterial colonies have not been conducted.

Another finding by Lenhart et al. (2015) is that $N_2O$ emissions are related to respiration by a relatively constant factor. By applying this factor and, furthermore, assuming a fixed ratio between respiration and NPP, the authors utilised the global NPP data of Elbert et al. (2012) to obtain globally resolved $N_2O$ emissions by lichens and bryophytes. The reliability of global estimates derived from upscaling of small-scale measurements depends on the variation of the measured fluxes. The field measurements of NPP, which were extrapolated to the spatial scale of a biome by Elbert et al. (2012), vary by around two orders of magnitude. Measurements of $N_2O$ emissions by lichens and bryophytes, too, show considerable variation. Regarding biological soil crusts, several studies analyzed denitrification rates to be negligible (Johnson et al., 2007; Strauss et al., 2012), and $N_2O$ production was calculated to constitute only 3-4 % of the N fixation rate (Barger et al., 2013). Other studies, however, described high denitrification rates that either increased (Brankatschk et al., 2013) or decreased with advancing crust development (Abed et al., 2013). One possibility to increase the reliability of large-scale estimates of $N_2O$ emissions by lichens and bryophytes is the application of alternative, methodically different approaches.

For this reason, we apply here the process-based non-vascular vegetation model LiBry (Porada et al., 2013) to assess the contribution of these organisms to the global $N_2O$ budget. LiBry simulates photosynthesis, respiration and growth of lichens and bryophytes as a function of environmental conditions. To distinguish global patterns of productivity on the ground and in the canopy, the model represents these locations and their differing environmental conditions separately. We calculate respiration by lichens and bryophytes directly as a function of environmental conditions and we derive $N_2O$-emissions based on the simulated respiration. By doing this, we obtain physiologically driven and spatially resolved data on the $N_2O$-emissions by lichens and bryophytes at the global scale. Since we estimate respiration with LiBry, we do not need to make assumptions regarding the ratio of NPP to respiration, contrary to Lenhart et al. (2015). Furthermore, we quantify different sources of variation in $N_2O$ emissions and determine their relative importance.

## 2   Methods

The non-vascular vegetation model LiBry estimates global patterns of photosynthesis, respiration and net primary productivity of lichens and bryophytes (Porada et al., 2013). The model calculates these physiological processes as a function of climate and additional environmental conditions, which are provided in form of time series of global gridded maps. Photosynthesis in LiBry is determined by ambient levels of light, $CO_2$ and temperature according to the Farquhar-approach (Farquhar and von Caemmerer, 1982). Respiration is simulated as a function of temperature via a $Q_{10}$-relationship. Both processes also depend on the water status of the simulated lichens and bryophytes, which includes limitation of $CO_2$-diffusion at high water content. NPP is derived from the difference between gross photosynthesis and respiration. A unique feature of LiBry is that functional diversity of lichens and bryophytes is represented by a large number of artificial species, instead of being aggregated into one or a few average functional types. The advantage of this approach is that adaptation of the organisms to differing environmental

conditions is simulated in a more realistic way. Physiological processes such as photosynthesis and respiration are calculated separately for each artificial species. LiBry has been successfully applied to estimate global NPP by lichens and bryophytes (Porada et al., 2013) and other impacts of these organisms on global biogeochemical cycles (Porada et al., 2014, 2016b).

The model version presented here contains several extensions compared to the original version: First, an NPP-based weighting scheme was introduced, which assigns relative abundances to all artificial species that survive in a grid cell of the model in the steady state (Porada et al., 2016b). This allows to derive an average grid cell value of NPP based on the relative abundances of the simulated species in that cell. In the original version, grid cell NPP could only be predicted in form of a range of values, due to unknown abundances of the species. The average grid cell NPP is close to the upper end of the range of productivity values, since the most productive simulated species are assumed to be the most abundant ones. Secondly, a dynamic disturbance scheme was implemented, which replaces the equilibrium computation of surface coverage by a monthly update of coverage (Porada et al., 2016a). This makes the new model applicable to transient scenarios of climatic and environmental change, while the original model required the assumption of a steady state to compute coverage.

For this study, we run LiBry with an initial value of 3000 artificial species in each grid cell for a period of 600 years to reach steady state, with climatic fields and other forcing data from Porada et al. (2013). Our global estimates are based on average values over the last 50 years of the simulation. We evaluate the new version of LiBry in the same way as the original one (Porada et al., 2013), by comparing simulated NPP to field measurements on a biome basis.

LiBry does not include an explicit representation of processes that directly result in emission of nitrous oxide. However, it has been determined experimentally by Lenhart et al. (2015) that $N_2O$ emissions by lichens and bryophytes are related to their respiration by a conversion factor of 16 ng $N_2O\,(mg\,CO_2)^{-1}$. The conversion factor has a 90 % confidence interval of 11 to 21 ng $N_2O\,(mg\,CO_2)^{-1}$. Since LiBry explicitly calculates respiration by lichens and bryophytes, we derive $N_2O$ emissions from simulated respiration using the conversion factor of Lenhart et al. (2015).

The study by Lenhart et al. (2015) uses NPP of lichens and bryophytes, together with free-living cyanobacteria and algae, to estimate $N_2O$ emissions, since global upscaled data on respiration of these organisms are not available from Elbert et al. (2012). Thereby, Lenhart et al. (2015) assume a fixed ratio of respiration to NPP. To determine this ratio, they evaluate literature data, obtaining a rate of respiration relative to net photosynthesis of ∼49 % (Lenhart et al., 2015, Tab. S6). Since measurements have been made in the sunlight, but respiration continues in the dark, respiration is multiplied by a factor of 2, assuming a 12-hour day. This leads to an estimated ratio of respiration to NPP which is roughly 1 : 1. To evaluate LiBry further, we compute the ratio of respiration to NPP in LiBry to assess if the model is in agreement with these observations.

In the study of Lenhart et al. (2015), the substrate of the samples was removed to avoid biases resulting from $N_2O$ release by microbes in the substrate. Foliose and fruticose lichens as well as mosses were collected, which grew on soil, rocks, and epiphytically on trees and the authors found no variation in $N_2O$ emissions depending on the underlying substrate. Endolithic and crustose lichens were not included in that study, as for these growth forms the dry weight, which is needed for calculations, could not be determined in a reliable manner.

Variation in field measurements of $N_2O$ emissions may result from physiological differences between species, but also from variation in climatic conditions, which can be significant at small scale. To upscale emissions from point measurements to

the large scale, it is important to quantify the relative contributions of these different sources of variation. If, for instance, the variation between species regarding their $N_2O$ emissions was small, it would suffice to sample a low number of species to obtain an average emission for a certain climatic condition. LiBry can provide an indication of the relative importance of these sources of variation, since the model not only represents climate variability, but also simulates diverse physiological

strategies. Each grid cell of the model contains a range of surviving species at the end of the simulation and, consequently, shows a range of $N_2O$ emissions. LiBry does not simulate spatial variation in climatic conditions within a grid cell. However, by comparing average emission rates of grid cells from different climates it is possible to assess the relative importance of climatic conditions for variation in $N_2O$ emissions. We select five model grid cells from different ecosystem classes to analyse the relative importance of differences between species and climatic heterogeneity on variation in $N_2O$ emissions. It should be

pointed out that LiBry does not compute directly $N_2O$ emissions by lichens and bryophytes, but it derives them from simulated respiration through an empirical linear relationship. Hence, differences in the sensitivities of respiration and $N_2O$ emissions to climatic conditions may lead to uncertainties in our predicted effects of climate on $N_2O$ emissions.

## 3  Results

The global distribution of net primary productivity simulated by the updated version of LiBry is shown in Fig. 1. Productivity

by lichens and bryophytes is highest in forested regions and lowest in deserts and agricultural regions. Hence, the spatial pattern is mainly controlled by water availability, except for cropland. In LiBry, it is assumed that lichens and bryophytes only grow on the area fraction of a grid cell which is not occupied by crops. Therefore, on a grid cell basis, regions with a high fractional cover of cropland show low productivity by lichens and bryophytes, in spite of favourable climatic conditions. The high productivity in the humid tropics mainly results from epiphytic lichens and bryophytes in the canopy, while in the boreal

zone, the larger fraction of productivity stems from the ground.

As a result of the dynamic surface coverage, the spatial patterns of NPP differ slightly between the new and the original version of LiBry, but the large scale gradients remain the same. Comparing the global pattern of lichen and bryophyte NPP simulated by the new version of LiBry to an empirical estimate by Elbert et al. (2012) shows good agreement, similar to the original version. Furthermore, the total global NPP predicted by the new LiBry differs from the original estimate due to the

updated calculation of coverage. The main difference is found for the tropical forest canopy, where simulated NPP increases significantly. The total global NPP of 4.3 $(Gt\,C)\,yr^{-1}$ estimated by the new LiBry compares well to the value of 3.9 $(Gt\,C)\,yr^{-1}$ calculated by Elbert et al. (2012).

Comparison of simulated NPP to field measurements on a biome basis suggests that LiBry predicts realistic values of NPP for a range of ecosystems (Fig. 2). In particular, simulated NPP in the tropical and the boreal forest matches well with

observations, while the original version of LiBry seemed to underestimate NPP in these biomes. In the biomes desert and, to a lesser extent, tundra, LiBry seems to overestimate productivity, which may have also been the case with the original version. A potential explanation for this is that productivity in dry and cold areas may not only be limited by climatic factors, but also by nutrient availability (Porada et al., 2016b). Since photosynthesis and growth are only controlled by climatic factors in LiBry,

the effect of spatial variation in nutrient availability on productivity cannot yet be simulated. It should be pointed out however, that, except for the boreal biome, the number of field measurements is quite low and, consequently, the observation-based characteristic values for each biome are subject to considerable uncertainty.

Figure 3 shows simulated global patterns of nitrous oxide emissions by lichens and bryophytes. Nitrous oxide emission is highest in the humid tropics and subtropics with values up to 10 $(mg\,N_2O)\,m^{-2}\,yr^{-1}$ (Fig. 3 a). A second region of high emissions is the boreal zone with values up to 8 $(mg\,N_2O)\,m^{-2}\,yr^{-1}$. Dry regions show lowest values of nitrous oxide emissions, in general less than 1 $(mg\,N_2O)\,m^{-2}\,yr^{-1}$. Considering only lichens and bryophytes which grow as epiphytes in the canopy (Fig. 3 b), emissions in the humid tropics are around three times higher than in the boreal and temperate zones. Lichens and bryophytes on the ground show highest values of nitrous oxide emissions in the boreal zone, with values around 3 $(mg\,N_2O)\,m^{-2}\,yr^{-1}$ (Fig. 3 c). Regarding the ground, tropical and subtropical regions only partly show $N_2O$ emissions comparable to those of the boreal zone. The reason for this is low simulated productivity and coverage of lichens and bryophytes on the ground in tropical and subtropical climates, which also leads to low respiration on a grid cell level and hence to low $N_2O$ emissions.

Figure 4 shows the simulated global spatial distribution of the ratio of respiration to NPP. The assumption of a globally constant ratio of respiration to NPP is used by Lenhart et al. (2015) to derive ecosystem-scale $N_2O$ emissions by lichens and bryophytes from their NPP. Alternatively, this ratio can be derived from the independent LiBry estimates of NPP and respiration. The simulated ratio shows a latitudinal pattern with increasing values towards the tropics (Fig. 4 a). This results from the influence of surface temperature on respiration in combination with high nighttime temperatures in the humid tropics, which cause high respiration rates during the night. Note that high respiration relative to NPP of tropical lichens and bryophytes does not necessarily mean high respiration at the grid cell level, since net productivity and coverage may be low. Respiration by lichens and bryophytes in the canopy shows a slightly weaker latitudinal gradient than the ground, which can be explained by efficient evaporative cooling in the canopy (Fig. 4 b). In contrast, lichens and bryophytes on the ground usually grow within the surface boundary layer, which reduces cooling by turbulent heat transfer, leading to a strong influence of incoming radiation on surface temperature. Since radiation input increases toward the equator, the ratio of respiration to NPP on the ground in the tropics is markedly higher than at high latitudes (Fig. 4 c). The ratio of respiration to NPP varies from less than 1 to around 2, while most values are around 1. This means that gross primary productivity (GPP) is partitioned roughly equally into NPP and respiration, which agrees well with the observational data from Lenhart et al. (2015).

An overview of global total values of $N_2O$ emissions, respiration, NPP and the ratio of respiration to NPP estimated by LiBry is shown in Tab. 1. Table 2 shows $N_2O$ emissions by lichens and bryophytes for individual grid cells from five different ecosystem classes (see also Tab. 1). Variation in emissions between species within a grid cell is large, it can exceed three orders of magnitude. The variation due to climatic conditions is smaller, but it still amounts to almost two orders of magnitude based on the grid cells with the highest and lowest average emission rates. Comparing Tab. 2 to the global range of $N_2O$ emissions by lichens and bryophytes (Fig. 3) shows that the five selected grid cells represent well the global variation in emissions due to climatic conditions. Thus, both functional diversity of the artificial species and different climatic conditions are important for variation of $N_2O$ emissions, according to the LiBry simulation.

## 4 Discussion

In this study we estimate nitrous oxide emissions by lichens and bryophytes with the global, process-based non-vascular vegetation model LiBry. Thereby, we derive $N_2O$ emissions from respiration fluxes which are, together with photosynthesis and net primary productivity, simulated by LiBry.

We use an updated version of LiBry which contains significant modifications with regard to the original version published in Porada et al. (2013). Regarding NPP, the new version estimates 4.3 $(Gt C) yr^{-1}$ while the original version of LiBry predicted a range of 0.34 to 3.3 $(Gt C) yr^{-1}$. The increase in predicted NPP is mainly attributed to a higher simulated productivity in the tropical forest canopy, since a new disturbance scheme allows for a higher surface coverage of lichens and bryophytes there. An empirical global estimate of NPP by lichens, bryophytes, free-living terrestrial cyanobacteria and algae (Elbert

et al., 2012) amounts to 3.9 $(Gt C) yr^{-1}$. Our new estimate is higher than that by Elbert et al. (2012), although LiBry does not consider free-living cyanobacteria and algae. This may be explained by the small contribution of cyanobacteria and algae to the overall global carbon uptake, which can be compensated by minor relative changes in productivity of lichens and bryophytes (Darrouzet-Nardi et al., 2015; Sancho et al., 2016). It is not straightforward to determine which number is closest to reality, since both the process-based estimate by LiBry as well as the empirical one by Elbert et al. (2012) are subject to uncertainty.

In the study of Elbert et al. (2012), for instance, it is assumed that productivity and active time are uniform within a biome. Furthermore, Elbert et al. (2012) use a globally uniform value of surface cover fraction to scale up local field measurements of productivity to the global scale. However, values of surface coverage by lichens and bryophytes compiled by Elbert et al. (2012) vary largely at the small scale, which makes upscaling to larger scales challenging.

     While productivity estimated by LiBry is evaluated in this study, large-scale surface coverage of lichens and bryophytes

simulated by LiBry has been evaluated for regions north of $50°$ N in Porada et al. (2016a). It was shown that LiBry predicts realistic values of cover fraction. Moreover, values of surface cover predicted by LiBry for other regions of the world (Porada et al., 2016b) are in agreement with the estimate of Elbert et al. (2012). In spite of uncertainties regarding productivity and abundance of lichens and bryophytes, comparing the empirical and process-based approaches gives confidence in the order of magnitude of the LiBry simulation results.

As a 50-year steady-state average value, we estimate total $N_2O$ emissions by lichens and bryophytes of 0.27 (0.19 - 0.35) $(Tg N_2O) yr^{-1}$, which is at the lower end of the range of 0.32 to 0.59 $(Tg N_2O) yr^{-1}$ calculated by Lenhart et al. (2015). The evaluation of LiBry regarding simulated NPP shows that our global patterns and total values of NPP are very similar to the empirical estimate by Elbert et al. (2012). Since Lenhart et al. (2015) use this NPP estimate by Elbert et al. (2012) to derive $N_2O$ emissions, differences in NPP are most likely not the reason for our lower estimate of $N_2O$ emissions compared to Lenhart

et al. (2015). Instead, this may be explained by differing methods to compute respiration: While Lenhart et al. (2015) assume a globally uniform ratio of respiration to NPP of the value 2 to estimate respiration, LiBry simulates respiration independently as a species-specific function of temperature and water status. This results in a lower global average value of around 1 for the ratio of respiration to NPP predicted by LiBry. Our estimated ratio of respiration to NPP agrees well with laboratory measurements, but it is in general difficult to compare a global, ecosystem-scale value to small-scale and short-term observations.

Our simulated global pattern of $N_2O$ emissions is slightly different than that shown in Lenhart et al. (2015), who estimate highest values in the boreal zone and only intermediate values in the humid tropics. This can be explained by their assumed constant ratio of respiration to NPP, which makes their global pattern of $N_2O$ emissions identical to that of NPP, which is shown in Elbert et al. (2012). In LiBry, however, the simulated ratio of respiration to NPP increases towards higher surface temperatures in the tropics (Fig. 4). Furthermore, it can be seen that the ratio shows large spatial variation. Evaluating this simulated pattern is difficult, since estimates which are extrapolated to the large scale, such as the NPP estimate by Elbert et al. (2012), are not available for respiration by lichens and bryophytes. However, observed ratios of respiration to NPP of lichens and bryophytes vary considerably at the species level, as shown by e.g. Lenhart et al. (2015). Using a constant ratio of respiration to NPP may therefore introduce a bias in the estimated spatial distribution of $N_2O$ emissions.

Small-scale measurements of $N_2O$ emissions by lichens and bryophytes may show considerable variation. The sources of this variation may be physiological differences between species, variation of associated microbial communities, as well as heterogeneity in climatic conditions. We examine the relative importance for respiration of differences between species compared to climatic differences with LiBry, since the model simulates various physiological strategies and represents variation in climatic conditions at the global scale. Thereby, we assume that the relationship between respiration and $N_2O$ emissions is relatively insensitive to climatic conditions and physiological differences between species, as suggested by the experiments by Lenhart et al. (2015). Table 2 shows that both differences between artificial species as well as different climatic conditions are important for variation of $N_2O$ emissions. Upscaling of $N_2O$ emission rates measured in the field may therefore be subject to considerable uncertainty. Modelling approaches in this direction should probably account for both interspecific variation in processes associated with $N_2O$ release by lichens and bryophytes as well as variation in climatic conditions.

Although our approach considers the most important sources of variation in $N_2O$ emissions by lichens and bryophytes, it is associated with uncertainties that should be discussed further. These uncertainties mainly result from our method to estimate respiration and from assumptions concerning the empirical relationship between respiration and $N_2O$ emissions.

Respiration and the ratio of respiration to NPP simulated by LiBry are difficult to validate, since the number of laboratory or field studies which measure not only NPP, but also GPP and respiration is not very high. Moreover, long-term measurements of respiration would be required to determine the ratio of respiration to NPP. Otherwise, assumptions about the contribution of respiration in the dark to total respiration are necessary.

To obtain $N_2O$ emissions from respiration, our results rely on the laboratory incubation measurements and the calculated ratio of $N_2O$ emissions to respiration presented in Lenhart et al. (2015). Furthermore, our approach considers effects of variation in climatic conditions on $N_2O$ emissions by lichens and bryophytes. Hence, it is necessary to discuss the sensitivity of the relationship between respiration and $N_2O$ emissions to a range of climatic conditions. As shown in Lenhart et al. (2015, Fig. 3), the relationship between respiration and $N_2O$ emissions seems to be insensitive to temperature changes for the tested species. Likewise, variations in water content have no clear effect on the relationship between $N_2O$ release and respiration (Lenhart et al., 2015, Fig. S3). Although the sensitivities of $N_2O$ release to temperature and water content are similar to those of respiration across species, the relationship between $N_2O$ release and respiration shows interspecific variation. However, in spite of a large number of around 40 sampled species, the relationship shows a relatively narrow 90 % confidence interval of

11.3 to 20.7 ng $N_2O\,(mg\,CO_2)^{-1}$ (Lenhart et al., 2015). This suggests that the mechanism of $N_2O$ release by lichens and bryophytes is similar between different species.

To analyse the relation between the production of $N_2O$ and respiratory $CO_2$ in greater detail, measurements of both fluxes by means of online gas exchange measurements would be needed, which then could be linked to the observed water status of the organisms. Since LiBry explicitly represents the dynamic water saturation of lichens and bryophytes, this would allow a more process-based prediction of the duration and magnitude of $N_2O$ emissions. In this way, the uncertainty associated with our approach would be reduced, facilitating an improved estimate of global $N_2O$ emissions by lichens and bryophytes.

In order to assess model-based estimates of $N_2O$ emissions by lichens and bryophytes, a relatively large number of field measurements are necessary. Currently, most $N_2O$ measurements, independently of the substrate or organisms measured, generally suffer from major uncertainties, additionally to variation from functional diversity and differing climatic conditions: first, the majority of these studies have been conducted using the acetylene inhibition technique. The idea of this method is to inhibit the last denitrification step, so that the measured $N_2O$-amounts should reveal the sum of $N_2O$ and $N_2$ release during denitrification under natural conditions. It has, however, been shown quite a while ago that this method leads to an underestimation of denitrification under oxic conditions (Bollmann and Conrad, 1997). Secondly, the most widely used measuring technique has been the closed chamber method, which is inexpensive and easy to use. This, however, has major shortcomings, as environmental conditions are hard to control and only limited surface areas can be measured (Butterbach-Bahl et al., 2013; Groffman, 2012). Furthermore, the limited temporal resolution of chamber measurements may affect estimated $N_2O$ emissions (Barton et al., 2015). Thirdly, depending on the environmental conditions under which the experiment is performed, particularly water, temperature, and nutrient conditions, the obtained $N_2O$ emission rates could differ widely. Thus, it is indispensable to report and consider the exact environmental conditions under which the measurements were made and to restrict natural emission data to those assessed under typically occurring natural conditions.

Respiration by lichens and bryophytes is not the only process which can be used to estimate their $N_2O$ emissions. Barger et al. (2013) report a relationship between nitrogen fixation and $N_2O$ release in biological soil crusts, which include lichens and bryophytes, but also soil bacteria and algae. For this approach, however, reliable nitrogen fixation data are sparse. It is also possible to estimate the demand for nitrogen by lichens and bryophytes with LiBry with an uncertainty range of around one order of magnitude (Porada et al., 2014). However, it is not straightforward to derive realised nitrogen uptake or nitrogen fixation from this, since LiBry does not yet include processes related to nitrogen uptake or metabolisation of nitrogen species. Therefore, for this study, we chose the relation between respiration and $N_2O$ release to quantify $N_2O$ emissions by lichens and bryophytes.

Our simulated global $N_2O$ emissions by lichens and bryophytes of 0.27 (0.19 - 0.35) $(Tg\,N_2O)\,yr^{-1}$ amount to around 3 % of global $N_2O$ emissions from natural sources on land (Ciais et al., 2013). This value may sound low at first glance, but it equals about 50 % of the atmospheric deposition of $N_2O$ into the oceans or 25 % of the deposition on land (Ciais et al., 2013). Considering that $N_2O$ has a strong negative effect on stratospheric ozone and a significant warming potential as a greenhouse gas, also relatively small emissions should not be neglected in global budgets.

The study by Zhuang et al. (2012) estimates global patterns of $N_2O$ emission from soils and finds that the humid tropics contribute most to global $N_2O$ emission due to high temperature and precipitation. Our simulated pattern of global $N_2O$ emissions by lichens and bryophytes also shows a hotspot in the humid tropics, but the relative contribution of the boreal zone to the global flux seems to be higher than in Zhuang et al. (2012). This probably results from the high simulated NPP in the boreal zone, particularly on the ground, which compensates for the lower respiration and therefore $N_2O$ emission per productivity due to low temperatures. Relative contributions of lichens and bryophytes to $N_2O$ emissions are highest for ecosystems in desert regions and at high latitudes, which agrees with the results by Lenhart et al. (2015).

## 5 Conclusions

We estimate large-scale spatial patterns and global values of $N_2O$ emissions by lichens and bryophytes from a process-based model of their productivity and respiration. Our results suggest a significant contribution of lichens and bryophytes to global $N_2O$ emissions, albeit at the lower end of the range of a previous, empirical estimate. Since both approaches use respiration to derive $N_2O$ emissions, our lower estimate likely results from a different method to predict respiration, compared to the empirical approach. Hence, while estimates of productivity are relatively well constrained, evaluating models with regard to estimated respiration may improve predictions of $N_2O$ emissions by lichens and bryophytes. One important finding derived from our simulation is that the ratio of respiration to NPP by lichens and bryophytes shows spatial variation and a latitudinal gradient at the global scale. This means that productivity and $N_2O$ emissions by the organisms are not necessarily correlated and that tropical regions may show higher emissions than polar regions given the same NPP. Furthermore, we show that both physiological variation among species as well as variation in climatic conditions are relevant for variation in respiration and, consequently, $N_2O$ emissions. Ecosystem-scale estimates of $N_2O$ emissions by lichens and bryophytes should therefore include sufficient ranges of species and climatic conditions to avoid biased results. Our results build on the empirical finding that $N_2O$ emissions by lichens and bryophytes are linearly related to their respiration. This relationship is relatively insensitive to climatic conditions and shows no large variation between species. However, the relationship is based on closed chamber measurements. Therefore, it would be useful to perform online gas exchange measurements of $N_2O$ emissions and respiration to test the effect of climatic conditions on the relationship between $N_2O$ release and respiration. Furthermore, using alternative approaches to estimate $N_2O$ emissions by lichens and bryophytes may be helpful to constrain our approach.

## 6 Code availability

The non-vascular vegetation model LiBry used here is combined with an interface for parallel computing which was developed at the Max Planck Institute for Biogeochemistry, Jena, Germany. LiBry without the interface is freely available as long as the copyright holders and a disclaimer are distributed along with the code in source or binary form. The code is available from the corresponding author upon request.

## 7   Data availability

Model output data which are presented as maps in this study are available as netCDF files from the authors on request.

*Author contributions.*   P. Porada, B.Weber and A. Kleidon designed the model simulations and P. Porada carried them out. P. Porada prepared the manuscript with contributions from all co-authors.

5   *Competing interests.*   The authors declare that they have no conflict of interest.

*Acknowledgements.*   This work has been supported by the PAGE21 project, grant agreement number 282700, funded by the EC seventh Framework Programme theme FP7-ENV-2011, and the CARBOPERM project, grant agreement number 03G0836B, funded by the BMBF (German Ministry for Science and Education). The authors thank the Max Planck Society (Nobel Laureate Fellowship for BW) for financial support. The Max Planck Institute for Biogeochemistry provided computational resources.

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

| | N$_2$O-emissions (Tg N$_2$O) yr$^{-1}$ | | NPP (Gt C) yr$^{-1}$ | Respiration (Gt C) yr$^{-1}$ | Respiration : NPP [ ] |
|---|---|---|---|---|---|
| Canopy + ground | | | | | |
| Global | 0.27 | (0.19 - 0.35) | 4.3 | 4.5 | 1.10 |
| Tropical forest | 0.11 | (0.08 - 0.14) | 1.5 | 1.8 | 1.33 |
| Extratropical forest | 0.11 | (0.08 - 0.14) | 2.0 | 1.8 | 0.93 |
| Steppe & Savannah | 0.03 | (0.02 - 0.04) | 0.4 | 0.4 | 1.21 |
| Desert | 0.02 | (0.01 - 0.03) | 0.4 | 0.4 | 1.05 |
| Tundra | 0.01 | (0.007 - 0.013) | 0.2 | 0.2 | 0.87 |
| Canopy, global | 0.13 | (0.09 - 0.17) | 2.1 | 2.2 | 1.01 |
| Ground, global | 0.14 | (0.10 - 0.18) | 2.2 | 2.3 | 1.16 |

**Table 1.** Annual global total values of N$_2$O emissions, NPP, respiration and the ratio of respiration and NPP estimated by LiBry and separated into lichens and bryophytes living in the canopy and on the ground. The values in brackets in the first column show the uncertainty in N$_2$O emissions due to the conversion of released CO$_2$ to N$_2$O (90 % confidence interval from Lenhart et al. (2015)). Ecosystem classes shown are based on the categories made by Olson et al. (2001), which were aggregated by us in the same way as in Elbert et al. (2012). "Gt C" stands for gigatons of carbon.

| Ecosystem class | Location | | Minimum | Average | Maximum |
|---|---|---|---|---|---|
| Tropical forest | Central Amazon | ground | 0.31 | 0.59 | 0.88 |
| | | canopy | 0.081 | 3.3 | 8.2 |
| Extratropical forest | West Siberia | ground | 0.023 | 1.7 | 4.8 |
| | | canopy | 0.0040 | 2.1 | 6.2 |
| Steppe & Savannah | Central Sahel | | 0.0095 | 0.088 | 0.32 |
| Desert | Central Australia | | 0.019 | 1.6 | 5.5 |
| Tundra | North Alaska | | 0.012 | 0.095 | 0.17 |

**Table 2.** Simulated nitrous oxide emissions by lichens and bryophytes in [(mg N$_2$O) m$^{-2}$ yr$^{-1}$] for individual grid cells of the LiBry model. The values are averages over the last 50 years of a 600-year simulation with 3000 initial species. Grid cells are selected from five different ecosystem classes. In the two forest classes, emissions are separated into canopy and ground. In the other classes, the model does not represent lichens and bryophytes in the canopy. The range of N$_2$O emissions based on all surviving artificial species in a grid cell is shown. The average value for all species in a grid cell is derived by an NPP-based weighting scheme (see Sect. 2).

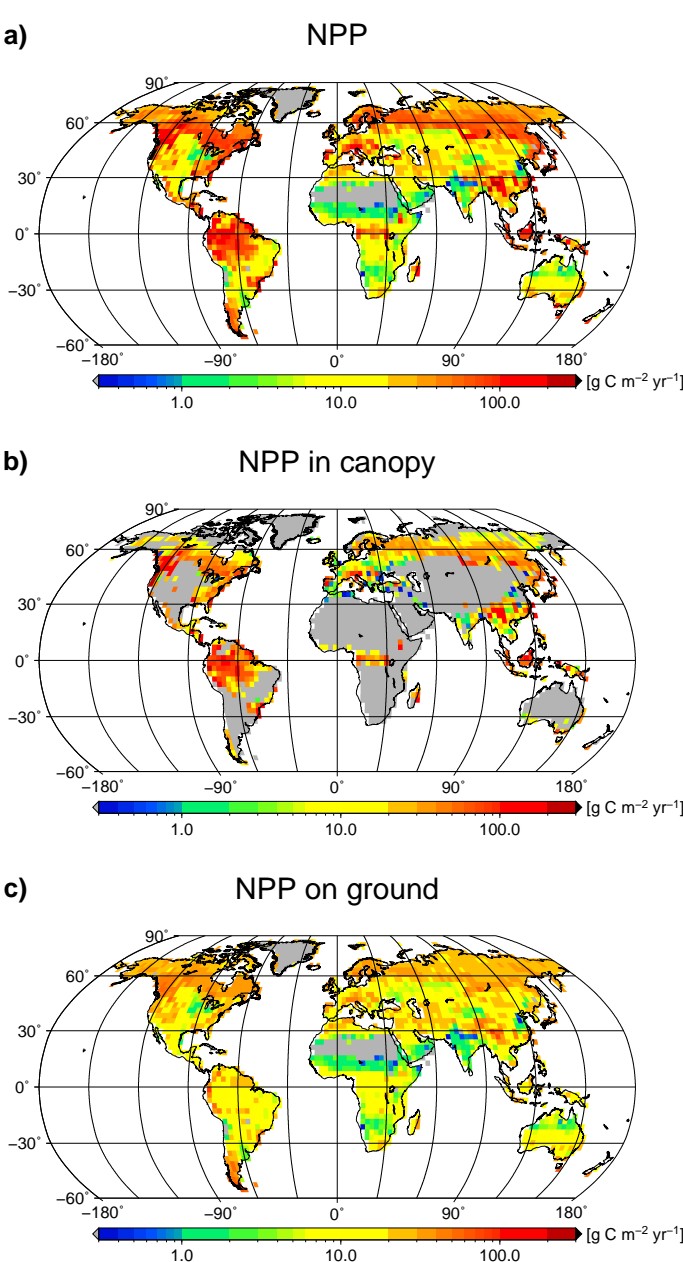

**Figure 1.** Global patterns of NPP. Lichen and bryophyte NPP estimated by LiBry for a) all locations of growth, b) the canopy and c) the ground. The estimates are in grams of carbon per m$^2$ and they are average values over the last 50 years of a 600-year simulation with 3000 initial species. Grey colour denotes regions where no simulated species is able to survive, such as ice shields and the driest regions of deserts.

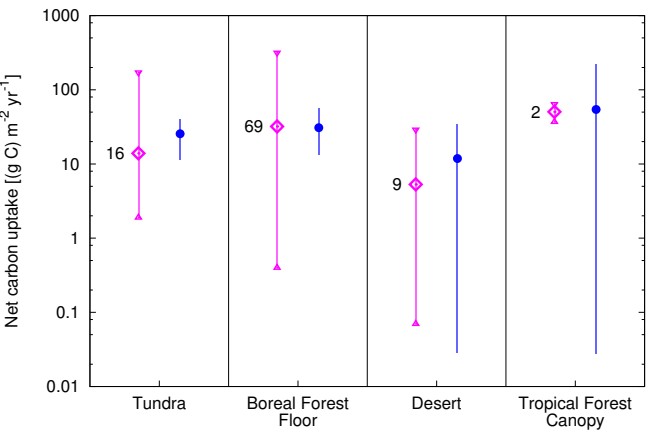

**Figure 2.** Comparison of LiBry estimates to field measurements. NPP estimated by LiBry compared to field measurements from four biomes, defined after Olson et al. (2001). The blue dots show the average simulated NPP for each biome and the blue vertical bars show the range of NPP values between the different grid cells in a biome. The magenta diamonds correspond to the median of NPP values measured in the field on the small scale, the magenta vertical bars denote the range of the field measurements. Left to the magenta diamonds the number of field measurements is shown that is considered for the respective biome. Details can be found in Porada et al. (2013).

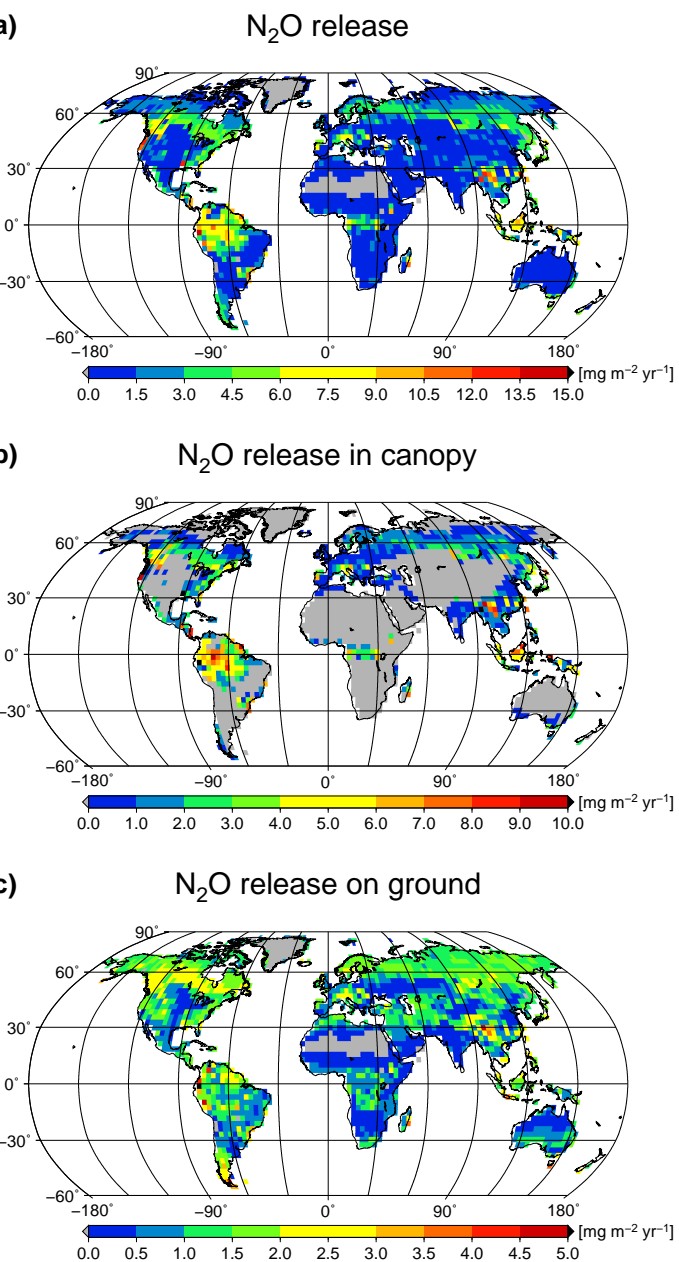

**Figure 3.** Global patterns of $N_2O$-release. Nitrous oxide emissions by lichens and bryophytes estimated by LiBry for a) all locations of growth, b) the canopy and c) the ground. Note the differing ranges of the color bars. Grey colour denotes regions where no simulated species is able to survive, such as ice shields and the driest regions of deserts.

**a)** Ratio respiration to NPP

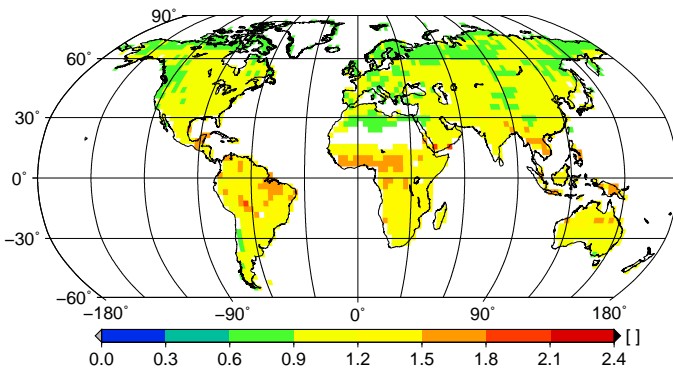

**b)** Ratio respiration to NPP in canopy

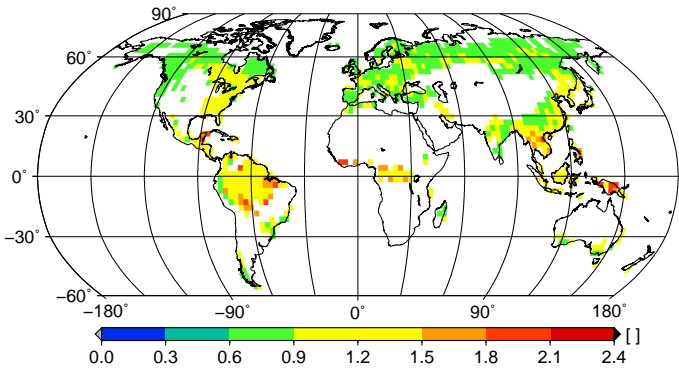

**c)** Ratio respiration to NPP on ground

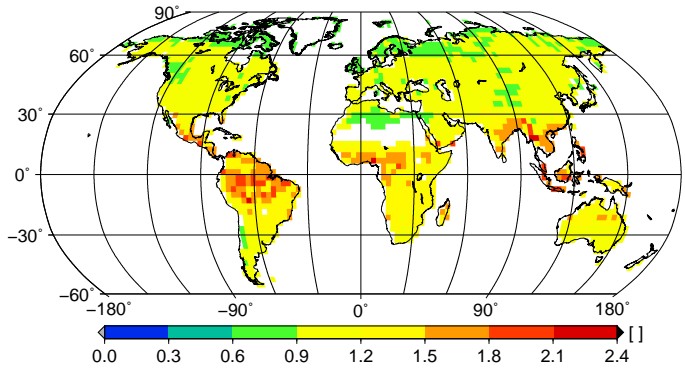

**Figure 4.** Global patterns of the ratio of respiration to NPP. Ratio of respiration to NPP of lichens and bryophytes estimated by LiBry for a) all locations of growth, b) the canopy and c) the ground.