# Peer review of "Estimating global nitrous oxide emissions by lichens and bryophytes with a process-based productivity model"

_Biogeosciences, 2016_

## Referee Comment (RC1) · Anonymous Referee #1 · 15 Nov 2016

Dear Dr. Porada and colleagues,

I have reviewed your manuscript "Estimating global nitrous oxide emissions by lichens and bryophytes with a process-based productivity model". In this manuscript, the updated model LiBry is used to estimate global respiration of lichens and bryophytes. Then global nitrous oxide emissions from lichens and mosses are derived from the simulated respiration amounts using a conversion factor. This is an important study, as the role of lichens and bryophytes in global biogeochemical cycles has been understudied. This is especially true for nitrous oxide, as exemplified by the fact that this paper is one of two global estimates for of N2O emissions for lichens and bryophytes. The model seems sound to me, and I appreciate the valid points the authors make

about the limitations of their emission estimates. However, there are some issues that need to be addressed within the manuscript.

General Comments

1. Conflation of mosses & bryophytes, biological soil crusts and microbial surface communities, and other terms- Please clarify if this paper is about one of these, all of these, or some of these. These terms are not interchangeable. The first paragraph of the introduction begins by talking about microbial surface communities (specifically biocrusts) in a dryland setting. However, the author's study is seeking to address global N2O emissions of lichens and mosses, as declared in the title. While lichens and mosses occur in biocrusts, the initial focus on biocrusts takes away from the global implications of the study and the potential importance of lichens and mosses to N2O emissions in other ecosystems (as the data later goes on to suggest). What is needed is less conflation of biocrusts with distinct units of lichens and mosses throughout the manuscript. This association of biocrusts with lichens and mosses is true for drylands, but the conflation breaks down very quickly in different ecosystems.

2. Clarify the players in N-cycling processes and the mechanisms early –This paper focuses on emissions of N2O actually sourced from the mosses and lichens themselves, not from nitrifiers or denitrifier microbes. Readers should be better introduced to this idea early on, so they are not confused. In the third, fourth, and fifth paragraph of the introduction, the focus is almost entirely on the fixation of N in microbial communities. These paragraphs are not entirely relevant to your study, and serve to confuse the reader.

Mechanisms for how microbial compounds release gaseous nitrogen are included, but there is no mention of the mechanisms for lichens and mosses until Page 9 Line 20. The process should be highlighted in the introduction. As follows, Figure 1 with its focus on the microbial communities mechanisms for N2O emissions is largely irrelevant to this study, and could be replaced by an example of lichen and mosses emission and

fixation pathways.

3. Make introduction global in scope - The results indicate that nitrous oxide emissions by lichens and bryophytes are highest in humid tropics and subtropics and yet, these regions are not even mentioned in the introduction. There is a lot of text spent on the N-dynamics of drylands, but I think it is more important to broaden the scope of the introduction and address the N-dynamics of the ecosystems that end up being most significant to global N20 emissions of lichens and mosses.

4. Expand the discussion: the discussion and conclusion focuses almost entirely on comparisons and short-comings of the model, while the introduction focuses heavily on N-cycling and mentions implications of N20 emissions. A paragraph tying the discussion back into the topics covered about lichens and mosses in the introduction, and our increased understanding of N20 emissions based on this study would be more satisfying to the reader. The authors begin to do this on page 9 line 15-18, but expanding on it or emphasizing it at the end of the manuscript would make for a stronger overall narrative.

Specific Comments

1. Page 4 line 8-16 Clarification of methods for relating N20 emissions with respiration –The explanation of how N2O emissions are derived from respiration states that they were converted from values determined experimentally from N2O emissions by microbial surface communities. It is important to note that lichens and mosses are not microbial surface communities. Mosses are plants! Neither are microscopic. I see later that it is stated Lenhart did measure samples of lichens and bryophytes. However, I had to read Lenhart et al to find that these measurements occurred when the lichens and mosses had their substrate (and therefore soil microbial communities) removed. Both of these points (1. Measurements were taken on lichens and bryophytes, not microbial surface communities and 2.removal of substrate during measurement) need to be made abundantly clear.

Also the morphological range of lichens and bryophytes used to get this conversion factor should be briefly mentioned. For instance, is respiration and N2O emissions as tightly coupled with rock lichen and epiphytes?

2. Page 9, line 12 The Elbert paper that is being cited includes cyanobacteria in its carbon estimates while this one does not. I would have guessed that the inclusion of cyanobacteria should make carbon estimates higher than carbon estimates from LiBry that focuses on just lichens and mosses. Please address this point.

3. Page 10 line 30-35. This paragraph is again conflating microbial surface communities with lichens and mosses. If the end goal is to assess model-based estimates of N2O emissions by microbial surface communities that contain lichens and mosses, then this paragraph is appropriate. However, that needs to be stated clearly.

Technical Points:

Page 6, line 12 "cannot not be simulated, yet" change to "cannot yet be simulated"

Page 10, line 17-18 Sentence fragment. Do you mean that the uncertainty you need to discuss involves the methods you used for estimating respiration and deriving N20 emissions from those respiration rates? If so, please state that more clearly.

---

## Referee Comment (RC2) · Anonymous Referee #2 · 15 Nov 2016

Review for "Estimating global nitrous oxide emissions by lichens and bryophytes with a process-based model" by Porada et al.

General comments:

The authors present a new approach to estimate global N2O emissions from lichens and bryophytes. In this approach they use empirical relationships between N2O and respiration to derive N2O emissions from simulated respiration fluxes. With this combination of modelling and empirical relationship they can represent the effect of climatic conditions on N2O emissions. Relating N2O emissions to climatic conditions is of course particularly interested in light of climate change. They highlight this, while they do not discuss that the sensitives in their N2O fluxes reflects the sensitivity of respiration. A more detailed discussion on potential differences in climatic sensitivities of N2O emissions vs respiration and related uncertainties is necessary. They discuss the advantage of their new approach vs previous estimates based on NPP and they also discuss shortcomings and general uncertainties related to N2O emissions by lichens and bryophytes. They state that their model does not simulate nitrification and denitrification, however, it does not get clear if the model is capable of simulating N fixation and N deposition. Those fluxes would have a more direct functional link to N2O emissions as compared to respiration. So in addition to referring to an alternative approach of using NPP, it would be beneficial to refer to other alternatives and related advantages or disadvantages of their approach. Another aspect still missing in the discussion is the general uncertainty related to estimates of the global abundance of lichens and bryophytes. With this extension of the discussion and the more specific comments below, I recommend the study for publication.

Specific comments:

Page 1:

Line 2 and 3: "This finding relies on . . . which are combined with . . .": It gets not very clear what the authors mean by "combined"; this is explained better later in the paper, but this sentence sounds too vague, please rephrase more clearly

Line 21: "In a first ecosystem-based upscaling approach": is this approach based on modelling or measuring on the ecosystem level? So is the alternative approach by Porada et al. (2013) different because they use a model (vs. observations) or because they model at global scale (vs. at ecosystem scale)?

Page 2:

Line 6: how can they influence weathering by their demand for phosphorous?

General remark: for those organisms fixing N, would it not make sense to link N2O emissions to fixed N? Or in general to N taken up, including fixed N; maybe this approach is not feasible in your case because of technical or modelling issues, but it would still be worth noting why you use respiration and not a N-related flux;

Line 8: are uptake into microbial biomass and leaching the only processes? Later you also mention gaseous losses, and your paper is about N2O emissions, so I guess you can expand this list. And is the uptake of fixed N relevant enough for the study for being dedicated one paragraph?

Line 11: how likely is it that nitrification and denitrification occur? As you derive global N2O emissions, do you distinguish between microbial communities that are and those that are not capable of nitrification or denitrification? If not, this fact should be discussed.

Line 12: what is meant by "surrounding atmosphere"? I suggest to delete "surrounding"

Line 13: ammonia is not formed during nitrification or denitrification

Line 17-19: who used those data?

Line 19: N2O is not in general the main ozone depleting substance, but the main ozone depleting substance that is still emitted; Other ozone depleting substances are not emitted any more, but still more destructive for ozone than N2O

Line 22 ff: in this paragraph you focus on denitrification, what about nitrification?

Line 22 ff: Regarding the upscaling of N2O emitted by lichens: how uncertain are estimates on global lichen and bryophyte occurrence?

Line 25: relation between N2O and fixation rate seems to be available from the study by Barger et al. 2013, why not using this relationship instead of linking N2O to respiration?

Page 3:

Figure 1: Figure 1 shows nitrification and denitrification, and the dependence of N2O emissions to NH4 and NO3 concentrations; It also shows that NH4 and NO3 depend

on fixation and deposition; In contrast to Figure 1, you derive N2O emissions from respiration; Is there a link between respiration and other N fluxes such as fixation? Is it pure coincidence that respiration and N2O fluxes show an empirical relationship?

Line 8 ff: ; is the relationship between N2O and respiration not driven by temperature change? Also moisture dependency of respiration might be different to N2O, especially as nitrification and denitrification have different optimum ranges; respiration differs between species... does N2O/respiration not differ across species? From what I found in cited literature, moisture dependency of respiration stays 1 for moisture values exceeding an upper limit; this is not true for N2O, as under very anoxic conditions, N2O is reduced further to N2: so here, the sensitivity of N2O on moisture differs from the one of respiration! This needs to be discussed at least.

Line 22: "...variation in climatic conditions": in the approach used in this study, the sensitivity of N2O emissions on climatic conditions mirrors the sensitivity of respiration; the authors do not discuss potential differences in sensitivities and arising uncertainties in their results, please add this to the discussion

Line 25: the variations in N2O emissions simulated in the study mirrors the variation in respiration; hence, claiming that their study helps to assess the variation in N2O emissions is a bit of a long shot; some clarification on this, and also on how reliable the linear relationship they are using is under different climatic conditions would be necessary

Page 5:

Line 4: "Since it is assumed in LiBry that lichens and bryophytes cannot grow together with crops, growth is low in these regions...": why do they grow at all, if it is stated that they cannot grow together with crops?

Figure 2: d) Tropical Forest Canopy: It seems like the small values come mainly from

very few grid cells at the edge of the tropics; if those few grid cells were excluded, range, and average value would look different; maybe I get this impression only due to the chosen color range, but I still think it wold be worth checking

Figure 2: what is the difference between organisms growing on ground or on leaves and how is this represented in the model? Here, that distinction comes up for the first time, if it is important to distinguish those two groups, then please add more explanation on it already in the introduction

Figure 2: values for desert regions are presented, while the Sahara is grey: please explain

Page 9:

The authors showed the ratio between respiration and NPP, however, they do not explain in how far respiration is dependent on NPP in the model; as N2O is somehow calculated from respiration, the link between N2O and NPP does not get clear; given this, the authors have a rather large focus on the NPP evaluation while it is not obvious how NPP affects N2O emissions in their approach

Page 10:

Line 9 ff: Diversity of estimated N2O emissions driven respiration, please add notes and discussions on that

Line 15: "functional diversity of lichens": I guess there are many kind of functional diversities and not all are related to N2O. . .. phrasing is a bit vague

Line 16: "considers the most important sources of variation. . .": this might be true for respiration, but you do not explicitly calculate N2O emissions, they mirror the sensitivity of respiration

Line 23 ff: one option to assess the uncertainty regarding wfps for N2O anyway could be to add a sensitivity of the linear relationship between N2O and respiration on water

content and test different ranges

Line 32: I assume not the measurements suffer from uncertainties, but that rather the results presented to not provide any information regarding uncertainties

Page 11:

Line 2: another shortcoming of manual chamber measurements is the limited temporal resolution which can make a huge difference in cumulated fluxes (Barton et al. 2015, Sampling frequency affects estimated of annual nitrous oxide fluxes, Scientific Reports)

Line 4: I don't really understand this sentence. How is water, temperature and nutrient conditions influenced by experimental setup? N2O emissions are driven by those factors, so it is quite logic that N2O emissions show a similar heterogeneity, independent of the experimental setup

Line 6: This sounds as if you refer to experiments with for instance application of fertilizer, that would in fact influence nutrient conditions by the experimental setup; if so then please phrase it more clearly

Conclusions: There are hardly any conclusions in the conclusion section; The first three sentences are a short summary of the study, the last sentence emphasizes vaguely how additional measurements could be beneficial; In my opinion you can draw more conclusions from your study, so please take a bit more care about this section. It is the last thing people read, and the way it reads now, it leaves at least me with an unsatisfied feeling about what actually your main conclusions are

Technical comments:

Line 8: units Tg N2O yr-1 or Tg N2O-N yr-1 ? – please specify the units regarding N2O emissions throughout the manuscript

Line 19: units: Gt C yr-1 or Gt CO2 yr-1 ?

Page 2:

Line 5: citation style

Line 17: N2O is already explained in line 13

Page 5:

Figure 2: units: change from [g m-2 yr-1] to [g C m-2 yr-1]

Page 9:

Line 15: again unit: Tg N or Tg N2O

Line 17: add blank after 25 %

---

## Author Comment (AC1) · 24 Jan 2017

**Estimating global nitrous oxide emissions by lichens and bryophytes with a process-based productivity model**

Philipp Porada[1,2], Ulrich Pöschl[3], Axel Kleidon[4], Christian Beer[1,2], and Bettina Weber[3]

[1]Department of Environmental Science and Analytical Chemistry (ACES), Stockholm University, 10691 Stockholm, Sweden
[2]Bolin Centre for Climate Research, Stockholm University, 10691 Stockholm, Sweden
[3]Max Planck Institute for Chemistry, P.O. Box 3060, 55020 Mainz, Germany
[4]Max Planck Institute for Biogeochemistry, P.O. Box 10 01 64, 07701 Jena, Germany

We thank the reviewer for useful and thorough comments which helped to improve our manuscript. We have prepared a revised manuscript where we account for all points raised by the reviewer, as described below. We show the reviewer's comments in italic text, while our responses are formatted as standard text.

**Response to the comments of reviewer #1**

*I have reviewed your manuscript "Estimating global nitrous oxide emissions by lichens and bryophytes with a process-based productivity model". In this manuscript, the updated model LiBry is used to estimate global respiration of lichens and bryophytes. Then global nitrous oxide emissions from lichens and mosses are derived from the simulated respiration amounts using a conversion factor. This is an important study, as the role of lichens and bryophytes in global biogeochemical cycles has been understudied. This is especially true for nitrous oxide, as exemplified by the fact that this paper is one of two global estimates for of N2O emissions for lichens and bryophytes. The model seems sound to me, and I appreciate the valid points the authors make about the limitations of their emission estimates. However, there are some issues that need to be addressed within the manuscript.*

We are glad that the reviewer appreciates the scientific relevance of our study and we have clarified all issues mentioned below.

*General Comments – 1. Conflation of mosses & bryophytes, biological soil crusts and microbial surface communities, and other terms – Please clarify if this paper is about one of these, all of these, or some of these. These terms are not interchangeable. The first paragraph of the introduction begins by talking about microbial surface communities*

*(specifically biocrusts) in a dryland setting. However, the authors study is seeking to address global N2O emissions of lichens and mosses, as declared in the title. While lichens and mosses occur in biocrusts, the initial focus on biocrusts takes away from the global implications of the study and the potential importance of lichens and mosses to N2O emissions in other ecosystems (as the data later goes on to suggest). What is needed is less conflation of biocrusts with distinct units of lichens and mosses throughout the manuscript. This association of biocrusts with lichens and mosses is true for drylands, but the conflation breaks down very quickly in different ecosystems.*

Our study focuses on $N_2O$ emissions by lichens and bryophytes. We agree with the reviewer that this should be made more clear in the introduction and we have therefore rephrased the respective parts in the revised manuscript. We have replaced the first paragraph by the following text:

"Lichens and bryophytes have increasingly been recognized to play a relevant role in global biogeochemical cycles [Elbert et al., 2012, Sancho et al., 2016, Barger et al., 2016]. They are globally abundant, growing on soils, rocks, and epiphytically on trees. At high latitudes, they may form extensive covers on the forest floor and in wetlands, mosses frequently represent the dominant vegetation type. In drylands, lichens and bryophytes form so-called biological soil crusts together with photosynthesizing cyanobacteria, algae, fungi and bacteria. These crusts cover vast areas in arid and semiarid ecosystems."

Throughout the manuscript we have exchanged the term "microbial surface communities" by "lichens and bryophytes" in case we refer to the paper by Lenhart et al. [2015], since their study describes measurements on lichens and bryophytes. When referring to the study by Elbert et al. [2012], we have replaced "microbial surface communities" by "lichens and bryophytes, together with free-living cyanobacteria and algae".

*2. Clarify the players in N-cycling processes and the mechanisms early – This paper focuses on emissions of N2O actually sourced from the mosses and lichens themselves, not from nitrifiers or denitrifier microbes. Readers should be better introduced to this idea early on, so they are not confused. In the third, fourth, and fifth paragraph of the introduction, the focus is almost entirely on the fixation of N in microbial communities. These paragraphs are not entirely relevant to your study, and serve to confuse the reader. Mechanisms for how microbial compounds release gaseous nitrogen are included, but there is no mention of the mechanisms for lichens and mosses until Page 9 Line 20. The process should be highlighted in the introduction. As follows, Figure 1 with its focus on the microbial communities mechanisms for N2O emissions is largely irrelevant to this study, and could be replaced by an example of lichen and mosses emission and fixation pathways.*

Already in the introductory part of the revised manuscript, we now discuss the potential processes responsible for $N_2O$ emissions from lichens and mosses. In fact, the underlying process causing $N_2O$ emissions from lichens and bryophytes is rather unclear. At the current stage there are two different ideas for this process. One option is that $N_2O$ could be directly released from the mosses and lichens in a similar way as it has been

described for plants [Smart and Bloom, 2001]. A second option is that bacteria growing on lichen and mosses are responsible for the emissions. This hypothesis is supported by a recent publication, where the bacterial species *Burkholderia* was isolated from leaves of the moss *Sphagnum fuscum* and was shown to emit $N_2O$ [Nie et al., 2015]. With the current situation of the emission process being not entirely clear, yet, we decided to do without a figure explaining the emission process.

To illustrate the relevance of lichens and bryophytes for global biogeochemical cycles, we have left the second and third paragraphs of the introduction largely unchanged, which describe fixation of $CO_2$ and nitrogen. We have then replaced paragraphs 4 to 6 with the following content:

[revised manuscript text omitted]

*3. Make introduction global in scope – The results indicate that nitrous oxide emissions by lichens and bryophytes are highest in humid tropics and subtropics and yet, these regions are not even mentioned in the introduction. There is a lot of text spent on the N-dynamics of drylands, but I think it is more important to broaden the scope of the introduction and address the N-dynamics of the ecosystems that end up being most significant to global N20 emissions of lichens and mosses.*

We have extended the introduction of the revised manuscript by a short overview of the relative contributions of lichens and bryophytes to nitrogen fluxes in various ecosystems (see previous point, paragraph 3 of the new text for the introduction).

*4. Expand the discussion: the discussion and conclusion focuses almost entirely on comparisons and short-comings of the model, while the introduction focuses heavily on N-cycling and mentions implications of N20 emissions. A paragraph tying the discussion back into the topics covered about lichens and mosses in the introduction, and our increased understanding of N20 emissions based on this study would be more satisfying to the reader. The authors begin to do this on page 9 line 15-18, but expanding on it or emphasizing it at the end of the manuscript would make for a stronger overall narrative.*

We have removed paragraph 3 and 6 from the discussion and instead added text on the global implications of our study and our understanding of $N_2O$ emissions by lichens and bryophytes at the end of the revised discussion. Moreover, we rephrased the conclusions (see below, reply to reviewer #2).

"Our simulated global $N_2O$ emissions by lichens and bryophytes of 0.27 (0.19 - 0.35) $(Tg\,N_2O)\,yr^{-1}$ amount to around 3 % of global $N_2O$ emissions from natural sources on land [Ciais et al., 2013]. This value may sound low at first glance, but it equals about 50 % of the atmospheric deposition of $N_2O$ into the oceans or 25 % of the deposition on land [Ciais et al., 2013]. Considering that $N_2O$ has a strong negative effect on stratospheric ozone and a significant warming potential as a greenhouse gas, also relatively small emissions should not be neglected in global budgets.

The study by Zhuang et al. [2012] estimates global patterns of $N_2O$ emission from soils and finds that the humid tropics contribute most to global $N_2O$ emission due to high temperature and precipitation. Our simulated pattern of global $N_2O$ emissions by lichens and bryophytes also shows a hotspot in the humid tropics, but the relative contribution of the boreal zone to the global flux seems to be higher than in Zhuang et al. [2012]. This probably results from the high simulated NPP in the boreal zone, particularly on the ground, which compensates for the lower respiration and therefore $N_2O$ emission per productivity due to low temperatures. Relative contributions of lichens and bryophytes to $N_2O$ emissions are highest for ecosystems in desert regions and at high latitudes, which agrees with the results by Lenhart et al. [2015]."

*Specific Comments – 1. Page 4 line 8-16 Clarification of methods for relating N20 emissions with respiration – The explanation of how N2O emissions are derived from respiration states that they were converted from values determined experimentally from N2O emissions by microbial surface communities. It is important to note that lichens and mosses are not microbial surface communities. Mosses are plants! Neither are microscopic. I see later that it is stated Lenhart did measure samples of lichens and bryophytes. However, I had to read Lenhart et al to find that these measurements occurred when the lichens and mosses had their substrate (and therefore soil microbial communities) removed. Both of these points (1. Measurements were taken on lichens and bryophytes, not microbial surface communities and 2.removal of substrate during measurement) need to be made abundantly clear. Also the morphological range of lichens and bryophytes used to get this conversion factor should be briefly mentioned. For instance, is respiration and N2O emissions as tightly coupled with rock lichen and epiphytes?*

We have made clear in the respective section that measurements were performed on lichens and bryophytes. Moreover, we have added a sentence regarding the removal of substrate: "...the substrate of the samples was removed to avoid biases resulting from $N_2O$ release by microbes in the substrate."

Nevertheless, we think that bacteria may well be involved in $N_2O$ emissions as explained above (reply to general comment 2.), paragraph 4 of the text newly inserted in the revised manuscript).

We also inserted information on the morphological range of the samples in the revised

manuscript: "Foliose and fruticose lichens as well as mosses were collected, which grew on soil, rocks, and epiphytically on trees and there was no variation in $N_2O$ emissions depending on the underlying substrate. Endolithic and crustose lichens were not included in that study, as for these growth forms the dry weight, which is needed for calculations, could not be determined in a reliable manner."

*2. Page 9, line 12 The Elbert paper that is being cited includes cyanobacteria in its carbon estimates while this one does not. I would have guessed that the inclusion of cyanobacteria should make carbon estimates higher than carbon estimates from LiBry that focuses on just lichens and mosses. Please address this point.*

We have added the following to the revised manuscript: "Our new estimate is higher than that by Elbert et al. [2012], although LiBry does not consider free-living cyanobacteria and algae. This may be explained by the small contribution of cyanobacteria and algae to the overall global carbon uptake, which can be compensated by minor relative changes in productivity of lichens and bryophytes."

*3. Page 10 line 30-35. This paragraph is again conflating microbial surface communities with lichens and mosses. If the end goal is to assess model-based estimates of N2O emissions by microbial surface communities that contain lichens and mosses, then this paragraph is appropriate. However, that needs to be stated clearly.*

We have changed the terms in the revised manuscript to be more clear.

*Technical Points: Page 6, line 12 cannot not be simulated, yet change to cannot yet be simulated Page 10, line 17-18 Sentence fragment. Do you mean that the uncertainty you need to discuss involves the methods you used for estimating respiration and deriving N20 emissions from those respiration rates? If so, please state that more clearly.*

We have made this sentence more precise: "These uncertainties result mainly from our method to estimate respiration and from assumptions concerning the empirical relationship between respiration and $N_2O$ emissions."

[revised manuscript text omitted]

---

## Author Comment (AC2) · 24 Jan 2017

**Estimating global nitrous oxide emissions by lichens and bryophytes with a process-based productivity model**

Philipp Porada[1,2], Ulrich Pöschl[3], Axel Kleidon[4], Christian Beer[1,2], and Bettina Weber[3]

[1]Department of Environmental Science and Analytical Chemistry (ACES), Stockholm University, 10691 Stockholm, Sweden
[2]Bolin Centre for Climate Research, Stockholm University, 10691 Stockholm, Sweden
[3]Max Planck Institute for Chemistry, P.O. Box 3060, 55020 Mainz, Germany
[4]Max Planck Institute for Biogeochemistry, P.O. Box 10 01 64, 07701 Jena, Germany

We thank the reviewer for useful and thorough comments which helped to improve our manuscript. We have prepared a revised manuscript where we account for all points raised by the reviewer, as described below. We show the reviewer's comments in italic text, while our responses are formatted as standard text.

**Response to the comments of reviewer #2**

*General comments: The authors present a new approach to estimate global N2O emissions from lichens and bryophytes. In this approach they use empirical relationships between N2O and respiration to derive N2O emissions from simulated respiration fluxes. With this combination of modelling and empirical relationship they can represent the effect of climatic conditions on N2O emissions. Relating N2O emissions to climatic conditions is of course particularly interested in light of climate change. They highlight this, while they do not discuss that the sensitives in their N2O fluxes reflects the sensitivity of respiration. A more detailed discussion on potential differences in climatic sensitivities of N2O emissions vs respiration and related uncertainties is necessary. They discuss the advantage of their new approach vs previous estimates based on NPP and they also discuss shortcomings and general uncertainties related to N2O emissions by lichens and bryophytes. They state that their model does not simulate nitrification and denitrification, however, it does not get clear if the model is capable of simulating N fixation and N deposition. Those fluxes would have a more direct functional link to N2O emissions as compared to respiration. So in addition to referring to an alternative approach of using NPP, it would be beneficial to refer to other alternatives and related advantages or disadvantages of their approach. Another aspect still missing in the discussion is the general uncertainty related to estimates of the global abundance of lichens and bryophytes. With this extension of the discussion and the more specific comments*

*below, I recommend the study for publication.*

In the revised manuscript, we point out that our findings regarding the effects of climate on $N_2O$ emissions depend on the climate sensitivity of the relation between $N_2O$ emissions and respiration. We explain that, so far, this relationship seems to be robust under a large range of environmental conditions, but we also mention that the detailed mechanisms of $N_2O$ emissions in lichens and bryophytes are still unclear. Furthermore, we discuss potential alternative approaches to derive $N_2O$ emissions as well as uncertainties regarding the global abundance of lichens and bryophytes.

*Specific comments*

*Page 1*
*Line 2 and 3: "This finding relies on ... which are combined with ...": It gets not very clear what the authors mean by "combined"; this is explained better later in the paper, but this sentence sounds too vague, please rephrase more clearly.*

We have replaced this sentence by: "This finding relies on ecosystem-scale estimates of net primary productivity of lichens and bryophytes, which are converted to nitrous oxide emissions by empirical relationships between productivity and respiration, as well as between respiration and nitrous oxide release."

*Line 21: "In a first ecosystem-based upscaling approach": is this approach based on modelling or measuring on the ecosystem level? So is the alternative approach by Porada et al. (2013) different because they use a model (vs. observations) or because they model at global scale (vs. at ecosystem scale)?*

We have clarified this section as follows: "In a first approach, based on empirical upscaling of field measurements according to ecosystem categories, Elbert et al. [2012] calculated that lichens and bryophytes, together with free-living cyanobacteria and algae, fix around $14.3\,(\mathrm{Gt\,CO_2})\,\mathrm{yr^{-1}}$ (3.9 Gt carbon) at the global scale. This corresponds to about $7\,\%$ of the net primary productivity (NPP) by terrestrial vegetation. As an alternative approach to the empirical upscaling of observations, Porada et al. [2013] utilized a process-based non-vascular vegetation model for lichens and bryophytes, called LiBry, to calculate the NPP of these organism groups at the global scale, obtaining similar results."

*Page 2*
*Line 6: how can they influence weathering by their demand for phosphorous?*

We have extended this point: "Moreover, it was found in the same study that the organisms may contribute significantly to biotic enhancement of global chemical weathering, by release of weathering agents such as organic acids. Their potential for chemical weathering was derived from their phosphorus demand, assuming that they dissolve surface

rocks to acquire phosphorus."

*General remark: for those organisms fixing N, would it not make sense to link N2O emissions to fixed N? Or in general to N taken up, including fixed N; maybe this approach is not feasible in your case because of technical or modelling issues, but it would still be worth noting why you use respiration and not a N-related flux;*

We explain in the discussion section of the revised manuscript why we do not use N uptake to derive rates of $N_2O$ emissions: "Respiration by lichens and bryophytes is not the only process which can be used to estimate their $N_2O$ emissions. Barger et al. [2013] report a relationship between nitrogen fixation and $N_2O$ release in biological soil crusts, which include lichens and bryophytes, but also soil bacteria and algae. It is possible to estimate the demand for nitrogen by lichens and bryophytes with LiBry with an uncertainty range of around one order of magnitude [Porada et al., 2014]. However, it is not straightforward to derive realised nitrogen uptake or nitrogen fixation from this, since LiBry does not yet include processes related to nitrogen uptake or metabolisation of nitrogen species. Therefore, for this study, we chose the relation between respiration and $N_2O$ release to quantify $N_2O$ emissions by lichens and bryophytes."

*Line 8: are uptake into microbial biomass and leaching the only processes? Later you also mention gaseous losses, and your paper is about N2O emissions, so I guess you can expand this list. And is the uptake of fixed N relevant enough for the study for being dedicated one paragraph?*

As described in our response to the comments of reviewer #1, we have rearranged this part of the introduction in the revised manuscript and we now focus on $N_2O$ emissions by lichens and bryophytes, their global significance and the associated metabolic processes. We removed those parts of the introduction which were not relevant for our approach, such as the description of various components of the nitrogen cycle in drylands.

*Line 11: how likely is it that nitrification and denitrification occur? As you derive global N2O emissions, do you distinguish between microbial communities that are and those that are not capable of nitrification or denitrification? If not, this fact should be discussed.*

As suggested by reviewer #1, we have clarified in the revised version of our manuscript that our estimate is constrained to $N_2O$ emissions by lichens and bryophytes. Moreover, we discuss the source of $N_2O$ emissions in greater detail: "...Lichens and bryophytes were shown to utilize $^{15}N$ labelled $NO_3^-$ but not $NH_4^+$, indicating that $N_2O$ is likely formed during denitrification. The exact process of $N_2O$-formation, however, remains largely unknown. One option is that the organisms themselves release $N_2O$ during the metabolisation of nitrate, in a similar way as suggested by Smart and Bloom [2001] for vascular plants. Another option is, that bacteria growing on lichen and moss cushions are responsible for the emissions of $N_2O$. This second option is supported by a recently

published study, where several strains of the bacterial genus *Burkholderia*, which were shown to emit $N_2O$, were isolated from the boreal peat moss *Sphagnum fuscum* [Nie et al., 2015]."

*Line 12: what is meant by "surrounding atmosphere"? I suggest to delete "surrounding"*

We have deleted this in the revised manuscript.

*Line 13: ammonia is not formed during nitrification or denitrification*

Due to the restructuring of the introduction in the revised manuscript, the corresponding paragraph has been deleted.

*Line 17-19: who used those data?*

We made clear in the revised manuscript that the data were used by Lenhart et al. [2015].

*Line 19: N2O is not in general the main ozone depleting substance, but the main ozone depleting substance that is still emitted; Other ozone depleting substances are not emitted any more, but still more destructive for ozone than N2O*

We extended this sentence to: "Since $N_2O$ is an important greenhouse gas and also the main depleting substance of stratospheric ozone which is still emitted today, ..."

*Line 22 ff: in this paragraph you focus on denitrification, what about nitrification?*

We point out in the revised manuscript that release of $N_2O$ by lichens and bryophytes is likely due to denitrification. Therefore, we focus in the respective paragraph on denitrification. As explained above, in the revised manuscript we have removed those parts of the introduction which describe processes not related to $N_2O$ emissions by lichens and bryophytes, but by soil organisms.

*Line 22 ff: Regarding the upscaling of N2O emitted by lichens: how uncertain are estimates on global lichen and bryophyte occurrence?*

We have added the following to the discussion: " ... In the study of Elbert et al. [2012], for instance, it is assumed that productivity and active time are uniform within a biome. Furthermore, Elbert et al. [2012] use a globally uniform value of surface cover fraction to scale up local field measurements of productivity to the global scale. However, values of surface coverage by lichens and bryophytes compiled by Elbert et al. [2012] vary largely at the small scale, which makes upscaling to larger scales challenging.

While productivity estimated by LiBry is evaluated in this study, large-scale surface coverage of lichens and bryophytes simulated by LiBry has been evaluated for regions north of 50° N in Porada et al. [2016a]. It was shown that LiBry predicts realistic values

of cover fraction. Moreover, values of surface cover predicted by LiBry for other regions of the world [Porada et al., 2016b] are in agreement with the estimate of Elbert et al. [2012]. In spite of uncertainties regarding productivity and abundance of lichens and bryophytes, comparing the empirical and process-based approaches gives confidence in the order of magnitude of the LiBry simulation results."

*Line 25: relation between N2O and fixation rate seems to be available from the study by Barger et al. 2013, why not using this relationship instead of linking N2O to respiration?*

As we explained above (Reply to "General remark", Page 2), we have added to the revised manuscript an explanation why we use respiration instead of N uptake to estimate $N_2O$ emissions.

*Page 3*
*Figure 1: Figure 1 shows nitrification and denitrification, and the dependence of N2O emissions to NH4 and NO3 concentrations; It also shows that NH4 and NO3 depend on fixation and deposition; In contrast to Figure 1, you derive N2O emissions from respiration; Is there a link between respiration and other N fluxes such as fixation? Is it pure coincidence that respiration and N2O fluxes show an empirical relationship?*

Unfortunately, the exact link between respiration and other N fluxes is not known, yet. Lenhart et al. [2015] worked with the empirical relationship between respiration and $N_2O$ fluxes and we adopted that for the current study. As suggested by us in the final discussion of the revised manuscript, "it would be useful to perform field measurements of $N_2O$ emissions and respiration to test the effect of climatic conditions on the relationship between $N_2O$ release and respiration. Furthermore, using alternative approaches to estimate $N_2O$ emissions by lichens and bryophytes may be helpful to constrain our approach. Integrating processes related to the metabolisation of nitrogen into our model may allow for a different method to quantify $N_2O$ release, independently of respiration."

*Page 4*
*Line 8 ff: ; is the relationship between N2O and respiration not driven by temperature change? Also moisture dependency of respiration might be different to N2O, especially as nitrification and denitrification have different optimum ranges; respiration differs between species. . . does N2O/respiration not differ across species? From what I found in cited literature, moisture dependency of respiration stays 1 for moisture values exceeding an upper limit; this is not true for N2O, as under very anoxic conditions, N2O is reduced further to N2: so here, the sensitivity of N2O on moisture differs from the one of respiration! This needs to be discussed at least.*

We have added the following to the discussion section of the revised manuscript:
"...our results rely on the laboratory incubation measurements and the calculated ratio of $N_2O$ emissions to respiration presented in Lenhart et al. [2015]. Furthermore, our approach considers effects of variation in climatic conditions on $N_2O$ emissions by

lichens and bryophytes. Hence, it is necessary to discuss the sensitivity of the relationship between respiration and $N_2O$ emissions to a range of climatic conditions. As shown in Lenhart et al. [2015, Fig. 3], the relationship between respiration and $N_2O$ emissions seems to be insensitive to temperature changes for the tested species. Likewise, variations in water content have no clear effect on the relationship between $N_2O$ release and respiration [Lenhart et al., 2015, Fig. S3]. Although the sensitivities of $N_2O$ release to temperature and water content are similar to those of respiration across species, the relationship between $N_2O$ release and respiration shows interspecific variation. However, in spite of a large number of around 40 sampled species, the relationship shows a relatively narrow 90 % confidence interval of 11.3 to 20.7 ng $N_2O$ (mg $CO_2$)$^{-1}$ [Lenhart et al., 2015]. This suggests that the mechanism of $N_2O$ release by lichens and bryophytes is similar between different species."

*Line 22: "...variation in climatic conditions": in the approach used in this study, the sensitivity of N2O emissions on climatic conditions mirrors the sensitivity of respiration; the authors do not discuss potential differences in sensitivities and arising uncertainties in their results, please add this to the discussion*

We added two sentences on the potentially different sensitivities of respiration and $N_2O$ emissions on climatic conditions to the methods section of the revised manuscript:

"It should be pointed out that LiBry does not compute directly $N_2O$ emissions by lichens and bryophytes, but it derives them from simulated respiration through an empirical linear relationship. Hence, differences in the sensitivities of respiration and $N_2O$ emissions to climatic conditions may lead to uncertainties in our predicted effects of climate on $N_2O$ emissions."

Moreover, we extended the discussion of the revised manuscript by a paragraph on the sensitivity of the relationship between respiration and $N_2O$ emissions to environmental conditions (see previous point).

*Line 25: the variations in N2O emissions simulated in the study mirrors the variation in respiration; hence, claiming that their study helps to assess the variation in N2O emissions is a bit of a long shot; some clarification on this, and also on how reliable the linear relationship they are using is under different climatic conditions would be necessary*

In the revised manuscript, we have pointed out potential effects of climatic conditions on the relationship between respiration and $N_2O$ emissions. We also have discussed these effects and their implications for our results (see previous two points).

*Page 5*
*Line 4: "Since it is assumed in LiBry that lichens and bryophytes cannot grow together with crops, growth is low in these regions ...": why do they grow at all, if it is stated that they cannot grow together with crops?*

In the revised manuscript, we have clarified this: "In LiBry, it is assumed that lichens

and bryophytes only grow on the area fraction of a grid cell which is not occupied by crops. Therefore, on a grid cell basis, regions with a high fractional cover of cropland show low productivity by lichens and bryophytes, in spite of favourable climatic conditions. "

*Figure 2: d) Tropical Forest Canopy: It seems like the small values come mainly from very few grid cells at the edge of the tropics; if those few grid cells were excluded, range, and average value would look different; maybe I get this impression only due to the chosen color range, but I still think it wold be worth checking*

It is true that the low values of productivity come from a few grid cells at the edge of the biome. However, it seems a bit arbitrary to us to change the boundary of the biome, which is derived from the map by Olson et al. [2001], to exclude some specific values. Since the number of grid cells per biome is very large (hundreds to thousands), excluding a few low values would not significantly shift the average value marked by the blue dot.

*Figure 2: what is the difference between organisms growing on ground or on leaves and how is this represented in the model? Here, that distinction comes up for the first time, if it is important to distinguish those two groups, then please add more explanation on it already in the introduction*

We added a short description of the different locations of growth simulated by LiBry to the introduction of the revised manuscript: "LiBry simulates photosynthesis, respiration and growth of lichens and bryophytes as a function of environmental conditions. To distinguish global patterns of productivity on the ground and in the canopy, the model represents these locations and their differing environmental conditions separately. "

*Figure 2: values for desert regions are presented, while the Sahara is grey: please explain*

In the revised manuscript, we have added to the caption of Fig. 2: "Grey colour denotes regions where no simulated species is able to survive, such as ice shields and the driest regions of deserts." Since we rearranged the figures in the revised manuscript, Fig. 2 is now Fig. 1

*Page 9: The authors showed the ratio between respiration and NPP, however, they do not explain in how far respiration is dependent on NPP in the model; as N2O is somehow calculated from respiration, the link between N2O and NPP does not get clear; given this, the authors have a rather large focus on the NPP evaluation while it is not obvious how NPP affects N2O emissions in their approach*

We explain in the methods section that NPP is calculated as the difference between photosynthesis and respiration and that respiration is calculated independently, as a function of temperature. To make this more clear, and to explain why we evaluate

NPP, we have extended and changed the respective section of the discussion in the revised manuscript: "...we estimate total $N_2O$ emissions by lichens and bryophytes of 0.27 (0.19 - 0.35) $(Tg\,N_2O)\,yr^{-1}$, which is at the lower end of the range of 0.32 to 0.59 $(Tg\,N_2O)\,yr^{-1}$ calculated by Lenhart et al. [2015]. The evaluation of LiBry regarding simulated NPP shows that our global patterns and total values of NPP are very similar to the empirical estimate by Elbert et al. [2012]. Since Lenhart et al. [2015] use this NPP estimate by Elbert et al. [2012] to derive $N_2O$ emissions, differences in NPP are most likely not the reason for our lower estimate of $N_2O$ emissions compared to Lenhart et al. [2015]. Instead, this may be explained by differing methods to compute respiration: While Lenhart et al. [2015] assume a globally uniform ratio of respiration to NPP of the value 2 to estimate respiration, LiBry simulates respiration independently as a species-specific function of temperature and water status. This results in a lower global average value of around 1 for the ratio of respiration to NPP predicted by LiBry. Our estimated ratio of respiration to NPP agrees well with laboratory measurements, but it is in general difficult to compare a global, ecosystem-scale value to small-scale and short-term observations. "

*Page 10*
*Line 9 ff: Diversity of estimated N2O emissions driven respiration, please add notes and discussions on that*

We have added the following to the revised manuscript: "...We examine the relative importance of these two factors with LiBry, since the model simulates various physiological strategies and represents variation in climatic conditions at the global scale. Thereby, we assume that the relationship between respiration and $N_2O$ emissions is relatively insensitive to climatic conditions and physiological differences between species, as suggested by the experiments by Lenhart et al. [2015]. "
As explained above, we have added a short discussion of potential effects of climatic conditions on the relationship between respiration and $N_2O$ emissions to the revised manuscript.

*Line 15: "functional diversity of lichens": I guess there are many kind of functional diversities and not all are related to N2O. . .. phrasing is a bit vague*

In the revised manuscript, we rephrased this sentence to: "Modelling approaches in this direction should probably account for both interspecific variation in processes associated with $N_2O$ release by lichens and bryophytes as well as variation in climatic conditions."

*Line 16: "considers the most important sources of variation. . .": this might be true for respiration, but you do not explicitly calculate N2O emissions, they mirror the sensitivity of respiration*

In the sentence following the quoted one, we refer to this potential uncertainty in our approach. To make this more clear, we have rephrased the sentence: "These uncertainties

result mainly from our method to estimate respiration and from assumptions concerning the empirical relationship between respiration and $N_2O$ emissions."

*Line 23 ff: one option to assess the uncertainty regarding wfps for N2O anyway could be to add a sensitivity of the linear relationship between N2O and respiration on water content and test different ranges*

As explained above, we have added a discussion on the sensitivity of the linear relationship to water content to the revised manuscript.

*Line 32: I assume not the measurements suffer from uncertainties, but that rather the results presented to not provide any information regarding uncertainties*

In the manuscript, our best estimate for $N_2O$ emissions is 0.27 $(Tg\,N_2O)\,yr^{-1}$. In addition to that, we give an uncertainty range, i.e., 0.19 - 0.35 $(Tg\,N_2O)\,yr^{-1}$.

*Page 11*
*Line 2: another shortcoming of manual chamber measurements is the limited temporal resolution which can make a huge difference in cumulated fluxes (Barton et al. 2015, Sampling frequency affects estimated of annual nitrous oxide fluxes, Scientific Reports)*

We have added this point to the revised manuscript: "...Furthermore, the limited temporal resolution of chamber measurements may affect estimated $N_2O$ emissions [Barton et al., 2015]."

*Line 4: I dont really understand this sentence. How is water, temperature and nutrient conditions influenced by experimental setup? N2O emissions are driven by those factors, so it is quite logic that N2O emissions show a similar heterogeneity, independent of the experimental setup*

We have replaced "experimental setup" by "environmental conditions under which the experiment is performed"

*Line 6: This sounds as if you refer to experiments with for instance application of fertilizer, that would in fact influence nutrient conditions by the experimental setup; if so then please phrase it more clearly*

With that statement, we had natural environmental conditions in mind. Hence, we state now: "...Thus, it is indispensable to report and consider the exact environmental conditions under which the measurements were made and to restrict natural emission data to those assessed under typically occurring natural conditions."

*Conclusions: There are hardly any conclusions in the conclusion section; The first three sentences are a short summary of the study, the last sentence emphasizes vaguely how*

*additional measurements could be beneficial; In my opinion you can draw more conclusions from your study, so please take a bit more care about this section. It is the last thing people read, and the way it reads now, it leaves at least me with an unsatisfied feeling about what actually your main conclusions are*

In the revised manuscript, we have rephrased the conclusions as follows: "We estimate large-scale spatial patterns and global values of $N_2O$ emissions by lichens and bryophytes from a process-based model of their productivity and respiration. Our results suggest a significant contribution of lichens and bryophytes to global $N_2O$ emissions, albeit at the lower end of the range of a previous, empirical estimate. Since both approaches use respiration to derive $N_2O$ emissions, our lower estimate likely results from a different method to predict respiration, compared to the empirical approach. Hence, while estimates of productivity are relatively well constrained, evaluating models with regard to estimated respiration may improve predictions of $N_2O$ emissions by lichens and bryophytes. One important finding derived from our simulation is that the ratio of respiration to NPP by lichens and bryophytes shows spatial variation and a latitudinal gradient at the global scale. This means that productivity and $N_2O$ emissions by the organisms are not necessarily correlated and that tropical regions may show higher emissions than polar regions given the same NPP. Furthermore, we show that both physiological variation among species as well as variation in climatic conditions are relevant for variation in respiration and, consequently, $N_2O$ emissions. Ecosystem-scale estimates of $N_2O$ emissions by lichens and bryophytes should therefore include sufficient ranges of species and climatic conditions to avoid biased results. Our results build on the empirical finding that $N_2O$ emissions by lichens and bryophytes are linearly related to their respiration. This relationship is relatively insensitive to climatic conditions and shows no large variation between species. However, the relationship is based on closed chamber measurements. Therefore, it would be useful to perform online flux measurements of $N_2O$ emissions and respiration to test the effect of climatic conditions on the relationship between $N_2O$ release and respiration. Furthermore, using alternative approaches to estimate $N_2O$ emissions by lichens and bryophytes may be helpful to constrain our approach."

*Technical comments*
*Line 8: units Tg N2O yr-1 or Tg N2O-N yr-1 ? - please specify the units regarding N2O emissions throughout the manuscript*

In the revised manuscript, we have replaced "Tg . . . of $N_2O$" or similar terms by "Tg $N_2O$".

*Line 19: units: Gt C yr-1 or Gt CO2 yr-1 ?*

In the revised manuscript, we have replaced "Gt . . . of carbon" or similar terms by "Gt C".

*Page 2*

*Line 5: citation style*

We have corrected this.

*Line 17: N2O is already explained in line 13*

We have corrected this.

*Page 5: Figure 2: units: change from [g m-2 yr-1] to [g C m-2 yr-1]*

We have changed this figure accordingly.

*Page 9: Line 15: again unit: Tg N or Tg N2O; Line 17: add blank after 25 %*

We have corrected this.

[revised manuscript text omitted]

---

## Referee Report (RR1)

After reviewing the revised manuscript, I feel the authors have sufficiently addressed the issues I raised in the initial review.  I applaud the thoroughness of the revision that the authors conducted in response to both reviewers' responses.

Minor issues:

Page 1, Line 2: Recently, lichens and bryophytes have also been shown
Page 1, Line 14: I am unsure of what you mean by "online flux measurements"
Page 2, line 7: In addition to photosynthetic carbon uptake,
Page 14, line 13:  Your explanation claiming that cyanobacteria and algae contribute small amounts to overall global carbon uptake needs a citation.  Possibly refer to *Darrouzet-Nardi, A., Reed, S.C., Grote, E.E., Belnap, J. (2015) Observations of net soil exchange of CO2 in a dryland show experimental warming increases carbon losses in biocrust soils.Biogeochemistry. 126:363-378*